# Geometrical Structure of Honeycomb TCP to Control Dental Pulp-Derived Cell Differentiation

**DOI:** 10.3390/ma13225155

**Published:** 2020-11-16

**Authors:** Kiyofumi Takabatake, Hidetsugu Tsujigiwa, Keisuke Nakano, Yasunori Inada, Shan Qiusheng, Hotaka Kawai, Shintaro Sukegawa, Shigeko Fushimi, Hitoshi Nagatsuka

**Affiliations:** 1Department of Oral Pathology and Medicine Graduate School of Medicine, Dentistry and Pharmaceutical Science, Okayama University, Okayama 700-8525, Japan; gmd422094@s.okayama-u.ac.jp (K.T.); keisuke1@okayama-u.ac.jp (K.N.); inayasu@s.okayama-u.ac.jp (Y.I.); hrbmushanqiusheng@163.com (S.Q.); de18018@s.okayama-u.ac.jp (H.K.); gouwan19@gmail.com (S.S.); fushimi@med.kawasaki-m.ac.jp (S.F.); jin@okayama-u.ac.jp (H.N.); 2Department of Life Science, Faculty of Science, Okayama University of Science, Okayama 700-0005, Japan; 3Department of Oral and Maxillofacial Surgery, Kagawa Prefectural Central Hospital, Kagawa 760-0065, Japan

**Keywords:** dental pulp, honeycomb TCP, matrix formation, dentin formation, geometrical structure, scaffold

## Abstract

Recently, dental pulp has been attracting attention as a promising source of multipotent mesenchymal stem cells (MSCs) for various clinical applications of regeneration fields. To date, we have succeeded in establishing rat dental pulp-derived cells showing the characteristics of odontoblasts under in vitro conditions. We named them Tooth matrix-forming, GFP rat-derived Cells (TGC). However, though TGC form massive dentin-like hard tissues under in vivo conditions, this does not lead to the induction of polar odontoblasts. Focusing on the importance of the geometrical structure of an artificial biomaterial to induce cell differentiation and hard tissue formation, we previously have succeeded in developing a new biomaterial, honeycomb tricalcium phosphate (TCP) scaffold with through-holes of various diameters. In this study, to induce polar odontoblasts, TGC were induced to form odontoblasts using honeycomb TCP that had various hole diameters (75, 300, and 500 μm) as a scaffold. The results showed that honeycomb TCP with 300-μm hole diameters (300TCP) differentiated TGC into polar odontoblasts that were DSP positive. Therefore, our study indicates that 300TCP is an appropriate artificial biomaterial for dentin regeneration.

## 1. Introduction

Functionally acting teeth are important for health and quality of life. Over the last few years, dental researchers have begun to explore methods for the repair and regeneration of dental structures by developing applications of stem cells and tissue engineering. Recently, the use of stem cells in molecular and cellular biology has led to the development of novel therapeutic strategies for regenerating oral tissues damaged by trauma or disease. Dental pulp is well known to be enriched with adult mesenchymal stem cells (MSCs), and stem cells isolated from dental pulp have high proliferation ability and might be able to differentiate into dentin-forming osteoblasts [1,2,3]. In addition, dental pulp stem cells play an important role in regenerative medicine, both for oral and non-oral region pathologies due to their high proliferation rates, multipotency, and accessibility [4,5]. Thus, dental pulp stem cells are a promising source of multipotent MSCs used in various clinical applications such as bone formation, dental tissue engineering, and the regeneration of neural tissue [6,7,8,9].

In our previous study, we established a steady dental pulp cell line derived from green fluorescent protein (GFP)-transgenic rats that shows dental pulp stem cell properties and stable odontoblastic differentiation both in vitro and in vivo. This rat dental pulp cell line has been maintained in culture for more than 80 passages without showing morphological changes or other properties [10]. We have named this cell line Tooth matrix-forming GFP rat-derived Cells (TGC). To our knowledge, there are no reports of established cells that stably exhibit osteoblastic characters and stem cell-like properties both in vitro and in vivo long-term. However, although this dental pulp cell line forms massive dentin-like hard tissues in vivo, it does not lead to the induction of polar odontoblasts.

A scaffold is an essential element for tissue engineering. Various artificial biomaterials have been developed as scaffolds and are already widely applied clinically. In recent years, some studies have focused on the biomaterial geometrical structure, because in addition to the composition, an optimal geometrical structure is considered important to induce cell proliferation and differentiation [11,12]. Focusing on this, we have already succeeded in developing a new biomaterial, honeycomb tricalcium phosphate (TCP) containing through-holes of various diameters. In our previous study, we reported that the surface properties resulting from the different sintering temperature affect the osteoinductivity and biocompatibility of TCP [13]. Furthermore, changing the through-holes diameters of honeycomb TCP holes allows for successful control of cartilage and bone formation [14]. In particular, in the skull defect model rat, vigorous bone tissue formation was observed in honeycomb TCP containing through-and-through holes with diameters of 300 μm, suggesting its clinical applicability [15]. These findings indicate that this honeycomb TCP can potentially act as a bioactive carrier and recapitulate the interactions between progenitor cells and the extracellular matrix microenvironment.

Therefore, the aim of this study was to generate polar odontoblasts using the cells we established and honeycomb TCP. The data presented here will help to improve existing dental tissue engineering methods via a stem cell-scaffold-based approach. In this study, to generate polar odontoblasts, TGC were differentiated into odontoblasts using honeycomb TCP as a scaffold. First, we conducted preliminary experiments ectopically using three types of honeycomb TCP, which has already been shown to form bone and cartilage tissue. Then, TGC and the optimal honeycomb TCP from the preliminary experiment were used to orthotopically induce odontoblasts.

## 2. Materials and Methods

### 2.1. Experiment Animals and Ethics

Six-week-old GFP transgenic female SD-Tg (CAG-EGFP) rats (SHIMIZU Laboratory Supplies, Co., Ltd., Kyoto, Japan) and twelve 4-week-old male severe combined immunodeficient (SCID) mice (CLEA Japan, Inc., Tokyo, Japan) were used in this study. The Animal Experiment Control Committee of Okayama University approved this study (OKU-2011048).

### 2.2. Cell Isolation and Culture

Pulp from mandibular incisors of GFP-transgenic rats was extracted and digested in a solution of 1 mg/mL collagenase type II (Invitrogen Co., New York, NY, USA) and 1 mg/mL dispase (Invitrogen Co., New York, NY, USA) at 37 °C for 2 h. Then single cell suspensions were cultured in Dulbecco’s minimal essential medium (DMEM; Invitrogen, Carlsbad, CA, USA) supplemented with 10% FBS and 100 U/mL antimycotic-antibiotic (Life Technologies, Thermo Fisher Scientific Inc., Tokyo, Japan). The cells were maintained by exchanging the medium every 3 days. Details of the isolation of TGC have been described previously [10].

Osteogenic medium used to evaluate the differentiation ability of TGC was made by adding ascorbic acid (50 μg/mL; Sigma-Aldrich, St. Louis, MO, USA) and glycerol 2-phosphate disodium salt hydrate (β-GP) (6 mM; Sigma-Aldrich) to the standard medium. Medium changes were made every 3 days.

### 2.3. Preparation of Honeycomb TCP Scaffolds

Honeycomb TCP used in ectopic experiments was pressed in a cylindrical mold containing through-and-through holes with diameters of 75 μm (75TCP), 300 μm (300TCP), and 500 μm (500TCP) (Figure 1a). Honeycomb TCP used in orthotopic experiments was pressed in a cylindrical mold containing through-and-through holes with diameters of 300 μm (Figure 1b). Each TCP was calcinated by heating to 1200 °C. Details of TCP manufacturing have been described previously [13].

### 2.4. Mineralization Assay by Alizarin Red Staining

TGC was washed once with phosphate-buffered saline (PBS) and fixed with 95% ethanol at 37 °C for 15 min. The fixed TGC was washed with distilled water (DW) and subsequently stained for 15 min with 1% alizarin red S (Katayama Chemical Industries, Co. Ltd., Osaka, Japan) solution. The remaining dye was washed out three times with DW. The stained samples were visualized by phase contrast microscopy using an inverted microscope.

### 2.5. Alkaline Phosphatase (ALP)

After TGC became confluence, culture medium was changed by osteogenic medium, or DMEM with 100 ng/mL recombinant human TGF-β (PeproTech, Rocky Hill, NJ, USA). The measurement of alkaline phosphatase activity was taken by p-Nitrophenyl Phosphatase Substrate method (FUJIFILM Wako Pure Chemical Co., Osaka, Japan), according to the manufacture’s instruction at 1, 3, 5 and 7 days.

### 2.6. Implantation and Histological Examination

SCID mice were anesthetized intraperitoneally with ketamine (75 mg/kg body weight) (Fuji Chemical Industry Co., Ltd., Tokyo, Japan) and Dormitol (0.5 mg/kg body weight) (Meiji Co., Ltd., Tokyo, Japan). 1 × 10^7^ TGCs were incubated in osteogenic medium or with TGF-β for 24 h, then TGCs were seeded onto the honeycomb TCP and implanted ectopically (subcutaneous back) or orthotopically (tibia). In ectopic experiments, the region of back was shaved, cleaned with 70% alcohol and iodine, and cut 5 mm by blunt dissection to form subcutaneous pockets. Each TCP was implanted carefully with tweezers in the subcutaneous pockets and sutured. In orthotopic experiments, using a 1.0-mm drill bit, a 3.0-mm hole was created at the center of the bone cortex approximately 5 mm away from the tibial epiphysis. At 4 weeks, the animals were euthanized with an overdose of isoflurane and were removed. All samples were fixed by 4% paraformaldehyde and decalcified with 10% ethylenediaminetetraacetic acid (EDTA). After decalcification, the samples were embedded in paraffin, sectioned at 5 μm in thickness, and stained by hematoxylin-eosin (H and E).

### 2.7. Immunohistochemical Staining of Dentin Sialprotein (DSP) and GFP

In this study, rabbit polyclonal anti-GFP (MBL, Nagoya, Japan) and rabbit polyclonal anti-DSP (Santa Cruz Biotechnology, Inc., Dallas, TX, USA) antibodies were used. The sections were deparaffinized in a series of xylene solutions for 15 min and then rehydrated, and incubated in 0.1% trypsin (Difco Laboratories, Detroit, MI, USA) for 5 min at 37 °C. Immunohistochemistry was performed using anti-GFP (dilution; 1:1000) or anti-DSP (dilution; 1:100) for 120 min at room temperature. Tagging of primary antibodies was achieved by the subsequent application of anti-goat secondary antibody by the use of the Histofine SAB-Po^®^ Kit (Nichirei, Tokyo, Japan) following the instruction of the manufacturer. Immunoreactivity was visualized using diaminobenzidine (DAB)/H_2_O_2_ solution (Histofine DAB substrate; Nichirei), and the slides were counterstained with Mayer’s hematoxylin.

### 2.8. Statistical Analysis

The statistical data was presented as the mean ± SEM. The comparisons between the mean variables of the two groups were performed using Student’s *t* test. The difference was considered significant at *p* < 0.05.

## 3. Results

### 3.1. Morphology of Isolated Cells

In culture, TGC showed a typical spindle-shaped fibroblast-like morphology with homogeneous shape and size (Figure 2a). Examination under a fluorescence microscope confirmed that the cells expressed GFP (Figure 2a). The TGC used in this experiment was the same as the cells in our previous study [10], therefore it was considered that the TGC retained their morphology and differentiation ability even after more than 80 passages without apparent change in their properties.

### 3.2. Differentiation Potential In Vitro

Alizarin red staining was used to quantify the mineral matrix depositions of TGC after 0, 2, 4, 7, 10, and 14 days of osteogenic induction culture. The mineral matrix deposition of TGC was found to increase over time until 14 days (Figure 2b).

Measurement of ALP of TGC after exposure to osteogenic medium or TGF-β showed an increase in ALP activity. The ALP activity from osteogenic medium became significantly higher than that from TGF-β at 3 and 5 days. After this point, the ALP activity of osteogenic medium-and TGF-β-exposed TGC decreased (Figure 2c).

### 3.3. Differentiation Potential In Vivo

First, to investigate the odontoblast induction ability of TGC and the honeycomb TCP complex, TGC and various pore-sized TCP were transplanted into SCID mice ectopically.

Bone-like hard tissue formation was observed near the entrance to the TCP holes in 75TCP. In 75TCP, hard tissue formation was observed as filling the pores (Figure 3a,b). In 300TCP, hard tissue formation was observed to be added to the TCP wall. In 300TCP, hard tissue formation around the TCP wall had polar dentin-like structure (Figure 3c,d). In 500TCP, hard tissue formation was observed to fill the pores or as added to the TCP wall (Figure 3e,f). The hard tissue formation in 75TCP and 500TCP did not have polar dentin-like structure.

To investigate the origin of hard tissue forming cells, we performed GFP immunohistochemical staining. The cells forming hard tissues in the TCP pores were GFP-positive, suggesting that these cells were derived from TGC (Figure 4a,c,e). DSP was not expressed in the cytoplasm of cells forming hard tissues in 75TCP and 500TCP (Figure 4b,e). In contrast, in 300TCP, DSP was expressed in the cytoplasm of cells that were arranged with polarity on the TCP wall, suggesting that they tended to differentiate into odontoblasts (Figure 4d).

To investigate orthotopically the odontoblast induction ability of TGC and the honeycomb TCP complex, which induced dentin-like structures in the ectopic experiment, complexes were transplanted into the tibias of SCID mice. In the TGF-β group, TGC showed massive hard tissue formation as if it was added to the wall of the honeycomb TCP. A polarity arrangement of TGC was observed around the massive hard tissue; however, most TGC were observed to be irregularly arranged (Figure 5a,b). TGC showed a tendency to differentiate into odontoblasts because they were DSP positive (Figure 5c). In the osteogenic medium group, we observed that TGC formed a highly columnar cell layer so as to be added to the honeycomb TCP wall, and showed polarity with nuclei arranged on the distal side of TCP (Figure 5d,e). TGC showed a tendency to differentiate into odontoblasts because they were DSP positive (Figure 5f).

## 4. Discussion

To the best of our knowledge, the present study is the first reported attempt to form polar dentin-like structures using artificial biomaterials and dental pulp stem cells.

Gronthos et al. first reported the isolation and characterization of dental pulp stem cells from pulp tissues of third molar impacted teeth and reported that dental pulp stem cells have greater cell proliferation and tissue regeneration abilities than bone marrow-derived mesenchymal stem cells [16]. Since then, many researchers have reported that dental pulp stem cells differentiate into various cells such as nerve cells, adipocytes, chondrocytes, and bone [17,18,19,20,21].

In our previous study, we established cells from GFP-transgenic rat dental pulp, and we have named these cells TGC (Tooth matrix-forming, GFP rat-derived Cells) [10]. TGC was able to differentiate into functional odontoblast-like cells both in vitro and in vivo. In vitro, exposure to an osteogenic medium resulted in an increase in ALP activity and the formation of calcium deposits (Figure 2b,c). As TGF-β is involved in dentin restoration and dentinogenesis [22,23], the effect of TGF-β on TGC was also investigated. The addition of TGF-β to TGC led to predictable results similar to those obtained with osteogenic medium. However, the stimulation of ALP activity by TGF-β was weaker than that resulting from exposure to osteogenic medium. One hypothesis to explain this weaker stimulation is that TGC can express proteins of the TGF family, and therefore, TGCs are already exposed to TGF-β.

For many tissues, especially hard tissues, regeneration using cells established from dental pulp and scaffolds has been reported. Polylactic acid (PLA), poly (α-hydroxyl) acids, polylactic-co-glycolic acid (PLGA), and ceramics as represented by TCP or Hydroxyapatite (HA) have been adapted as scaffolds for dentin regeneration [24,25,26]. It has already been verified that these artificial biomaterials have high biocompatibility, and they have been applied as scaffolds for inducing odontoblast differentiation as well as bone induction. Gronthos et al. first reported that transplanted dental pulp stem cells with an HA-TCP scaffold in mice induced the formation of odontoblast-like hard tissue [3]. To date, there have been some reports of dentin-like hard tissue formation in experiments combining various scaffolds and dental pulp stem cells [27,28]. Among these artificial biomaterials, TCP has particularly high biocompatibility, and it has been reported that when it is transplanted into a living body, it is absorbed over time and self-assembled, which is advantageous for hard tissue formation [29,30,31]. However, many studies using TCP and dental pulp cells to induce the differentiation of odontoblasts, and our previous experiments using granular TCP, have not led to the regeneration of polar dentin [10].

As described above, many studies have used artificial biomaterials suitable for inducing differentiation into odontoblasts. However, there is still no study that has effectively induced differentiation into odontoblasts by changing the geometric structure of the same artificial biomaterial. In this study, histological observation showed that the hard tissue formed in the pores was different when the pore sizes of honeycomb TCP were changed. In ectopic experiments, bone-like hard tissue formation was observed for 75TCP and 500TCP. However, for 300TCP, hard tissue formation was observed as if it was added to the TCP wall, and since these cells were DSP positive, they tended to differentiate into dentin (Figure 3 and Figure 4). Furthermore, they had a polar arrangement and exhibited odontoblast-like structures as if they were found in the pulp cavity. In our previous study, the angiogenesis regulated by the pore size of honeycomb TCP influenced on hard tissue formation [14]. The previous study indicated that the invasive blood vessel formation pattern in 300TCP which showed capillary pattern was different from that in 500TCP, which showed the same linear angiogenesis. It was possible that the capillaries induced by 300TCP mimicked the vascular construction in the dental pulp environment and could induce dentin-like structures of more than 500TCP, which induces thick linear blood vessels, and 75TCP which induces only thin blood vessels. In the orthotopic experiment, using 300TCP, hard tissue formation with polarity was observed as if added to the wall of TCP, and DSP was expressed (Figure 5). Since 300 μm is close to the thickness of the pulp [32,33], TCP of 300 μm imitates the pulp environment in vivo, and we consider that this is why dentin with such polarity was formed. Therefore, we think that the specific microenvironment provided by 300TCP is involved in odontoblast differentiation of TGC.

Although the mechanisms that control TGC differentiation into osteoblasts in vitro and in vivo are not fully understood at this time, it is clear that the inductive microenvironment consisting of proteins, genes, and factors, in addition to the scaffold and seeding cells, are necessary for dentin regeneration. Finally, the main limitation of this study was whether TGC works in human body. TCP has high biocompatibility and strength, however, in the future we would like to examine whether the dentin-like hard tissue formed in this experiment has the strength and biocompatibility to function physiologically in the living body.

## 5. Conclusions

Our study indicates that 300TCP is an appropriate artificial biomaterial for dentin regeneration, because 300TCP induced TGC to differentiate into polar odontoblast-like hard tissue that is similar to a normal dentin-like structure. Therefore, 300TCP may serve as a new biological material for dentin regeneration and a 300TCP/TGC complex could be used as a model for further studies about dentin regeneration.

## Figures and Tables

**Figure 1 materials-13-05155-f001:**
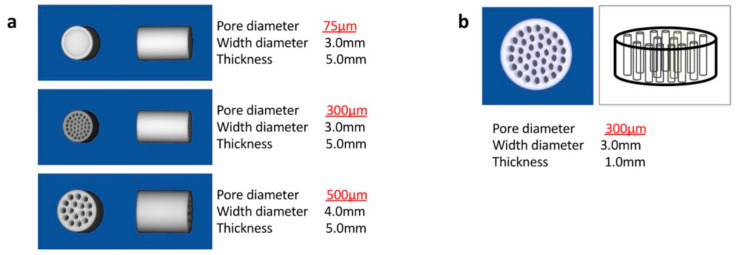
The honeycomb tricalcium phosphate (TCP) structures used in these experiments. (**a**) The honeycomb TCP used in the ectopic experiment. (**b**) The honeycomb TCP used in the orthotopic experiment.

**Figure 2 materials-13-05155-f002:**
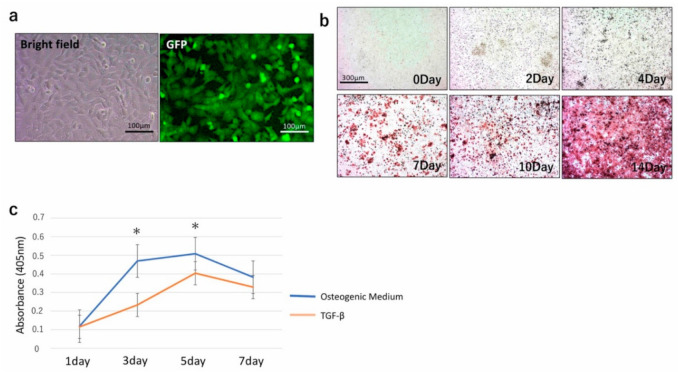
(**a**) TGC showed a fibroblastic shape (left), and fluorescent microscopy confirmed the expression of green fluorescent protein (GFP) (right). (**b**) Alizarin red staining of TGC exposed to osteogenic medium from 0 to 14 days. (**c**) Measurement of Alkaline Phosphatase (ALP) activity of TGC cultivated with osteogenic medium or TGF-β. The ALP activity from osteogenic medium became significantly higher than that from TGF-β at 3 and 5 days. After this point, the ALP activity of osteogenic medium- and TGF-β-exposed TGC decreased. * *p* < 0.05.

**Figure 3 materials-13-05155-f003:**
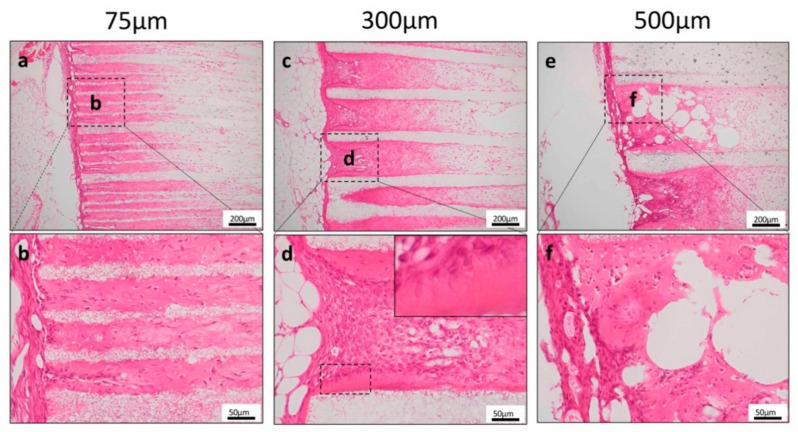
Histological findings in the ectopic experiment. (**a**,**b**) Bone-like hard tissue formation was observed near the entrance to the TCP holes for 75TCP. (**c**,**d**) Dentin-like structures around the TCP were observed in 300TCP. (**e**,**f**) Hard tissue formation was observed as added to the TCP wall in 500TCP.

**Figure 4 materials-13-05155-f004:**
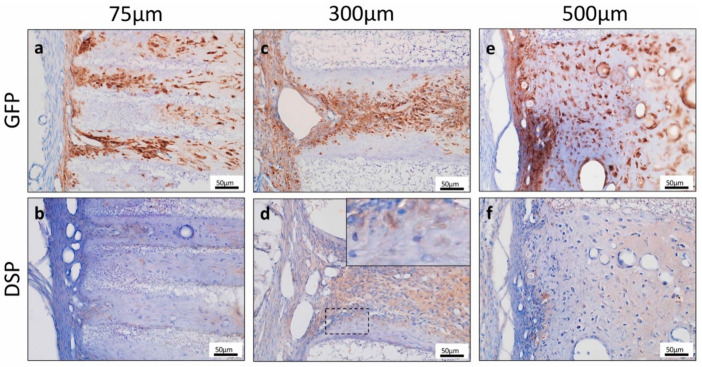
(**a**,**c**,**e**) The cells forming hard tissues in the TCP pores were GFP positive. (**b**,**d**,**f**) Dentin Sialprotein (DSP) was not expressed in the cytoplasm of cells forming hard tissues in 75TCP and 500TCP. In contrast, in 300TCP, DSP was expressed in the cytoplasm of cells that were arranged with polarity on the TCP wall.

**Figure 5 materials-13-05155-f005:**
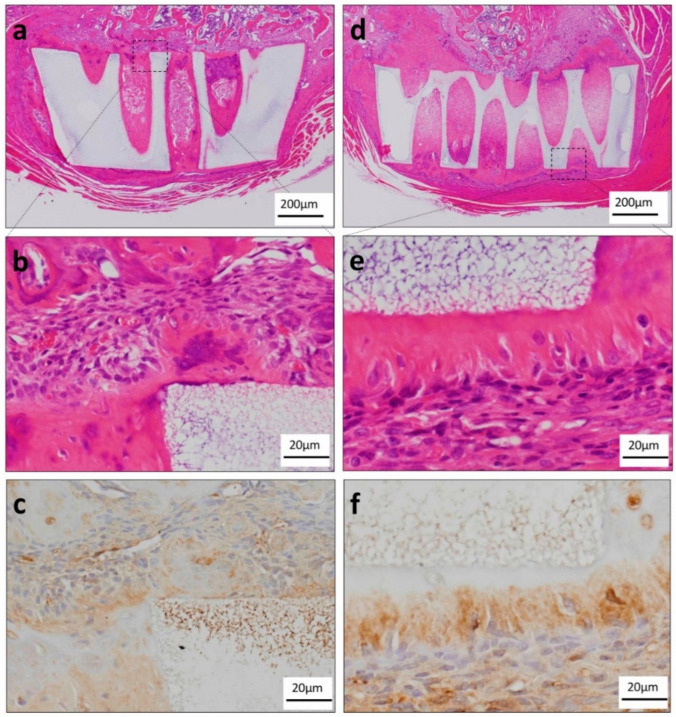
Histological findings in the orthotopic experiments. (**a**) Histological finding of the TGF-β group in low-power magnification. TGC showed massive hard tissue formation as if it was added to the wall of the honeycomb TCP. (**b**) Histological findings of the TGF-β group in high-power magnification. Most of the TGC were observed to be irregularly arranged. (**c**) TGC showed a tendency to differentiate into odontoblasts because they were DSP positive. (**d**) Histological finding of the osteogenic medium group in low-power magnification. TGC showed massive hard tissue formation as if it was added to the wall of the honeycomb TCP. (**e**) Histological finding of the osteogenic medium group in high-power magnification. TGC which differentiated into highly columnar cells formed a single layer hard tissue so as to be added to the honeycomb TCP wall, and showed polarity with the nuclei arranged on the distal side of TCP. (**f**) TGC tended to differentiate into odontoblasts because they were DSP positive.

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
