# Peer review of "Geometrical Structure of Honeycomb TCP to Control Dental Pulp-Derived Cell Differentiation"

_materials, 2020, doi:10.3390/ma13225155_

Round 1
Reviewer 1 Report
In this study Kiyofumi Takabatake and collaborators demonstrate that honeycomb TCP with 300-μm hole diameters is an appropriate artificial biomaterial for dentin regeneration.
Overall the paper is well written. However, the following issues should be addressed in this manuscript in order to be of interest for the readers of Materials:
- In the results paragraph authors assess “The TGC retained their morphology and differentiation ability even after more than 80 passages without apparent change in their properties” but these data are missed. Did the authors culture the cells for more than 80 passages? Please explain this and add results showing that cells did not change their properties at high passage. Also the cells at passage 80 were differentiated? Please show data of their differentiation ability.
- Figure 2: in figure 2C is the measurement of ALP activity of TGC cultivated with TGF-β statistically significant? The p value <0.05 is related to absorbance at 3 and 5 days compared to the absorbance at 1 day? Please make this legend clear.
- How many cells have been isolated from pulp mandibular incisors of GFP-transgenic rats?
- An image of both ectopically and orthotopically implantation in mice should be added in the manuscript.
- If possible to do, a quantification of massive hard tissue formation to the wall of the honeycomb TCP should be added.
- Please better explain the translational implication of this study in humans. Is the clinical application of 300TCP for the treatment in humans achievable? Also what is the limitation of this study? Please improve the discussion.
- As angiogenesis exerts an influence on bone tissue formation, please discuss this point.
- The same author have recently published the paper: “Effect of Honeycomb β -TCP Geometrical Structure on Bone Tissue Regeneration in Skull Defect”. I was wondering why the authors did not cite this paper in the manuscript.
Minor
- Please explain in the text the full meaning of DSP (dentin sialoprotein). This is never reported in the text.
- Figure 2a I would suggest to write “bright field” instead of “morphology”.
- Legend figure 5 d,e please better write this sentence because it is not clear.
Author Response
Thank you very much for providing important insights. We are delighted to hear that you think our work will spark debate in our field. In the following sections, you will find our responses to each of your points and suggestions. We are grateful for the time and energy you expended on our behalf.
- In the results paragraph authors assess “The TGC retained their morphology and differentiation ability even after more than 80 passages without apparent change in their properties” but these data are missed. Did the authors culture the cells for more than 80 passages? Please explain this and add results showing that cells did not change their properties at high passage. Also the cells at passage 80 were differentiated? Please show data of their differentiation ability.
→We did not examine the differentiation ability of TGC in this experiments. However, in this study, we used the same TGC in our previous study (reference 10). The TGC was passaged more than 80, and did not change their properties in our previous study. We have modified the text in the results (Line 153-155).
- Figure 2: in figure 2C is the measurement of ALP activity of TGC cultivated with TGF-β statistically significant? The p value <0.05 is related to absorbance at 3 and 5 days compared to the absorbance at 1 day? Please make this legend clear.
→We have modified the figure legend.
- How many cells have been isolated from pulp mandibular incisors of GFP-transgenic rats?
→We have strictly counted the number of cells required for transplantation, however, the number of cells at the time of primary culture was not counted.
- An image of both ectopically and orthotopically implantation in mice should be added in the manuscript.
→We have added to the detail protocols of ectopic or orthotopic implantation (Line 124-128).
- If possible to do, a quantification of massive hard tissue formation to the wall of the honeycomb TCP should be added.
→Thank you for your suggestion. We will consider to do in the future.
- Please better explain the translational implication of this study in humans. Is the clinical application of 300TCP for the treatment in humans achievable? Also what is the limitation of this study? Please improve the discussion.
→We have added to the texts about clinical application and the limitation of this study in Discussion (Line 289-292).
- As angiogenesis exerts an influence on bone tissue formation, please discuss this point.
→We have discussed about angiogenesis in Discussion (Line 274-280).
- The same author have recently published the paper: “Effect of Honeycomb β -TCP Geometrical Structure on Bone Tissue Regeneration in Skull Defect”. I was wondering why the authors did not cite this paper in the manuscript.
→We have mentioned about honeycomb TCP including our previous manuscript (skull defect model) in Introduction (Line 64-66).
Minor
- Please explain in the text the full meaning of DSP (dentin sialoprotein). This is never reported in the text.
→We have added to the abbreviation of DSP.
- Figure 2a I would suggest to write “bright field” instead of “morphology”.
→We have changed morphology to bright field.
- Legend figure 5 d,e please better write this sentence because it is not clear.
→We have modified the figure legend.
Reviewer 2 Report
Dear Editor
Please find enclosed my comments about the manuscript entitled: "Geometrical structure of honeycomb TCP to control dental pulp-derived cell differentiation".
The manuscript reports about a Tooth matrix-forming, GFP rat-derived Cells (TGC), which form massive dentin-like hard tissues under in vivo conditions, however this does not lead to the induction of polar odontoblasts. A honeycomb TCP scaffold with different pore diameters was used to induce polar odontoblasts. Interestingly they found that the honeycomb TCP with 300-μm hole diameters differentiated TGC into polar odontoblasts that were DSP positive suggesting that this biomaterial would be appropriate for dentin regeneration. I found the manuscript interesting and I think it may be of interest for the readers of the journal.
After reading the manuscript I have the following comments:
The quality of figure 2A should be improved. The microscopy images are bit blur and the letters inside the graph are too small. In all pictures (a and c) the scale bar is missing.
It is not clear how the authors characterized the TCP scaffolds. Especially the diameter of the pore, was it determined before or after calcination?
In my opinion the conclusion are little bit too brief and some more information can be incorporated.
Some text editing is needed along the manuscript. For example: "Figure 5. (a,b)In the... (c)TGC... etc."
Author Response
Thank you very much for providing important insights. We are delighted to hear that you think our work will spark debate in our field. In the following sections, you will find our responses to each of your points and suggestions. We are grateful for the time and energy you expended on our behalf.
- The quality of figure 2A should be improved. The microscopy images are bit blur and the letters inside the graph are too small. In all pictures (a and c) the scale bar is missing.
→We have modified the Figures.
- It is not clear how the authors characterized the TCP scaffolds. Especially the diameter of the pore, was it determined before or after calcination?
→We have already mentioned the characters of TCP in Materials and Methods. We determined the diameter of the pore size before calcination. The characters such as the crystal structure and surface properties of TCP after firing have mentioned in our previous study (reference 13).
- In my opinion the conclusion are little bit too brief and some more information can be incorporated.
→We have modified the conclusion by mentioning TGC (Line 306-310).
- Some text editing is needed along the manuscript. For example: "Figure 5. (a,b)In the... (c)TGC... etc."
→We have modified the sentences.